# β-Lapachone Exerts Anticancer Effects by Downregulating p53, Lys-Acetylated Proteins, TrkA, p38 MAPK, SOD1, Caspase-2, CD44 and NPM in Oxaliplatin-Resistant HCT116 Colorectal Cancer Cells

**DOI:** 10.3390/ijms24129867

**Published:** 2023-06-07

**Authors:** Eun Joo Jung, Hye Jung Kim, Sung Chul Shin, Gon Sup Kim, Jin-Myung Jung, Soon Chan Hong, Choong Won Kim, Won Sup Lee

**Affiliations:** 1Department of Internal Medicine, Institute of Health Sciences, Gyeongsang National University Hospital, Gyeongsang National University College of Medicine, 15 Jinju-daero 816 Beon-gil, Jinju 52727, Republic of Korea; eunjoojung@gnu.ac.kr; 2Department of Pharmacology, Institute of Health Sciences, Gyeongsang National University College of Medicine, Jinju 52727, Republic of Korea; hyejungkim@gnu.ac.kr; 3Department of Chemistry, Research Institute of Life Science, Gyeongsang National University, Jinju 52828, Republic of Korea; sshin@gnu.ac.kr; 4Research Institute of Life Science, College of Veterinary Medicine, Gyeongsang National University, Jinju 52828, Republic of Korea; gonskim@gnu.ac.kr; 5Department of Neurosurgery, Institute of Health Sciences, Gyeongsang National University Hospital, Gyeongsang National University College of Medicine, Jinju 52727, Republic of Korea; gnuhjjm@gnu.ac.kr; 6Department of Surgery, Institute of Health Sciences, Gyeongsang National University Hospital, Gyeongsang National University College of Medicine, Jinju 52727, Republic of Korea; hongsc@gnu.ac.kr; 7Department of Biochemistry, Institute of Health Sciences, Gyeongsang National University College of Medicine, Jinju 52727, Republic of Korea; cwkim@gnu.ac.kr

**Keywords:** β-lapachone, anticancer effect, oxaliplatin-resistant, colorectal cancer, p53, aggresomes, antibody array, chemotherapy

## Abstract

β-lapachone (β-Lap), a topoisomerase inhibitor, is a naturally occurring *ortho*-naphthoquinone phytochemical and is involved in drug resistance mechanisms. Oxaliplatin (OxPt) is a commonly used chemotherapeutic drug for metastatic colorectal cancer, and OxPt-induced drug resistance remains to be solved to increase chances of successful therapy. To reveal the novel role of β-Lap associated with OxPt resistance, 5 μM OxPt-resistant HCT116 cells (HCT116-OxPt-R) were generated and characterized via hematoxylin staining, a CCK-8 assay and Western blot analysis. HCT116-OxPt-R cells were shown to have OxPt-specific resistance, increased aggresomes, upregulated p53 and downregulated caspase-9 and XIAP. Through signaling explorer antibody array, nucleophosmin (NPM), CD37, Nkx-2.5, SOD1, H2B, calreticulin, p38 MAPK, caspase-2, cadherin-9, MMP23B, ACOT2, Lys-acetylated proteins, COL3A1, TrkA, MPS-1, CD44, ITGA5, claudin-3, parkin and ACTG2 were identified as OxPt-R-related proteins due to a more than two-fold alteration in protein status. Gene ontology analysis suggested that TrkA, Nkx-2.5 and SOD1 were related to certain aggresomes produced in HCT116-OxPt-R cells. Moreover, β-Lap exerted more cytotoxicity and morphological changes in HCT116-OxPt-R cells than in HCT116 cells through the downregulation of p53, Lys-acetylated proteins, TrkA, p38 MAPK, SOD1, caspase-2, CD44 and NPM. Our results indicate that β-Lap could be used as an alternative drug to overcome the upregulated p53-containing OxPt-R caused by various OxPt-containing chemotherapies.

## 1. Introduction

β-lapachone (β-Lap) is a naturally occurring *ortho*-naphthoquinone compound, originally isolated from the lapacho tree (*Tabebuia avellanedae*), and can be synthesized from lapachol, naphthoquinones and other aromatic compounds [1,2]. β-Lap is known to have variety of pharmacological activities, including anticancer, antioxidant, antifungal, antibacterial, anti-inflammatory, anti-obesity, neuroprotective and nephroprotective properties, with low toxicity, mainly through the activation of NAD(P)H: quinone oxidoreductase 1 (NQO1) and the inhibition of topoisomerase I [1,2,3]. The anticancer effect of β-Lap has been associated with the downregulation of mutant p53 in SW480 human colon cancer cells, and it has also occurred in human colon cancer cells (HCT116, wild-type p53; SW620, mutant p53; DLD1, defective p53) and human prostate cancer cells (LNCaP, wild-type p53; DU-145, mutant p53; PC-3, null-type p53), regardless of p53 status, by inducing cell cycle arrest and apoptosis [4,5,6]. In addition, p53 is known to be upregulated in the nucleus via co-treatment with β-Lap and paclitaxel, a diterpene taxane natural compound, in Y79 human retinoblastoma cells, leading to apoptosis through the downregulation of Akt phosphorylation; however, p53 protein levels were not significantly altered by β-Lap treatment or paclitaxel alone [7]. These results suggest that the anticancer effects of β-Lap could be affected by p53 protein status, depending on the intracellular circumstances.

Colorectal cancer is one of the most common malignant cancers worldwide. Drug resistance due to long-term chemotherapy is a major cause of tumor recurrence, metastasis and high mortality in various cancer treatments, so it is important to elucidate new mechanisms to enhance anticancer effects and develop new therapies to overcome drug resistance [8,9]. CD133 and CD44 are known as colorectal stem cell markers and are implicated in drug resistance to standard chemotherapy [10,11]. Oxaliplatin (OxPt) is a third-generation platinum analog of the 1,2-diaminocyclohexane families, inhibits DNA repair, replication and transcription through cross-linking with inter- and intra-strand DNA, and induces cell cycle arrest and apoptosis in various cancers [12,13]. OxPt-induced apoptosis is known to be related to an increase in phosphorylated p53 and p38 mitogen-activated protein kinase (p38 MAPK) and the induction of H2AX phosphorylation in serine 139 (γ-H2AX) in HCT116 colorectal cancer cells [14,15]. Since oxaliplatin resistance (OxPt-R) is a main cause of OxPt-containing chemotherapeutic failure in various cancers, many studies on OxPt-R-related mechanisms to overcome OxPt-R have been performed in OxPt-R patient samples and stable cell lines generated via long-term treatment with OxPt [16,17,18,19,20,21,22]. The following representative OxPt-R mechanisms are known: the inhibition of OxPt transport into cells through the downregulation of hCTR1; the activation of OxPt efflux through the upregulation of ATP-binding cassette (ABC) drug transporters; detoxification through the upregulation of GSH; DNA repair through the upregulation of ERCC1, XPF, XPD and DNA polymerase; the activation of NF-κB-mediated survival signaling; and the suppression of cell death signaling through the upregulation of survivin, MMP7 and autophagy [17,22]. Nonetheless, OxPt-R-related mechanisms are still unclear, so further studies are needed to improve the success of various OxPt-containing chemotherapies.

Previous studies have suggested that β-Lap could play an important role in suppressing drug resistance caused by long-term chemotherapy in various types of cancer through its potent anticancer activity [23,24]. In this study, we first generated and characterized 5 μM OxPt-resistant HCT116 cells (HCT116-OxPt-R) to reveal the novel role of β-Lap associated with OxPt-R in wild-type p53-containing HCT116 cells. Next, we identified new OxPt-R-related proteins through signaling explorer antibody array analysis using HCT116-OxPt-R cells. Finally, we compared the anticancer effect and molecular mechanism of β-Lap between HCT116 cells and HCT116-OxPt-R cells. According to the result, the anticancer effect of β-Lap was higher in HCT116-OxPt-R cells than in HCT116 cells, leading to more cytotoxicity and morphological changes in HCT116-OxPt-R cells. Moreover, the anticancer mechanism of β-Lap in HCT116-OxPt-R cells containing certain aggresomes and upregulated p53 was associated with the downregulation of OxPt-R-related proteins such as p53, survivin, ERK, Lys-acetylated proteins, TrkA, p38 MAPK, SOD1, caspase-2, CD44 and NPM. Thus, we suggest that β-Lap could be used as a natural chemotherapeutic agent with low cytotoxicity for the treatment of various OxPt-R cancers through the inhibition of multiple OxPt-R-related proteins, including upregulated p53.

## 2. Results

### 2.1. Generation of 5 μM OxPt-Resistant HCT116 Cells

To further elucidate OxPt-R properties in wild-type p53-containing colorectal cancer, we generated 5 μM OxPt-resistant HCT116 cells from HCT116 cells via long-term treatment with OxPt for several months. As shown in Figure 1a,b, OxPt-R cells were treated with a gradually increasing OxPt concentration from 0.2 μM to 1 μM such that the indicated passages would not to have significant morphological changes and cytotoxicity compared to parental HCT116 cells; the resulting 1 μM OxPt-R cells were stimulated with a high concentration of 6.3 μM OxPt for two passages to strongly induce OxPt properties; 6.3 μM OxPt-R cells were stabilized with a lower concentration of 2.5 μM OxPt for 10 passages; and 2.5 μM OxPt-R cells were treated with 5 μM OxPt for 16 passages. As a result, 5 μM OxPt-resistant stable HCT116 cells (HCT116-OxPt-R) were generated, and HCT116-OxPt-R cells at passage 16 (P16) were stored in a liquid nitrogen tank. Parental HCT116 cells and HCT116-OxPt-R cells (P16) were used in this study, and the results were confirmed through repeated experiments. First, we investigated the morphological differences between HCT116 and HCT116-OxPt-R cells from P17 to P22 via phase-contrast microscopy to find a novel OxPt-R-related phenotype. Notably, HCT116-OxPt-R cells appeared to be more complex than HCT116 cells, probably due to certain accumulated macromolecules produced through long-term treatment with OxPt (Figure 1c). This fact suggests that complex morphological changes in HCT116-OxPt-R cells may be associated with OxPt-R properties against OxPt-induced cytotoxicity.

### 2.2. Characterization of HCT116-OxPt-R Cells

#### 2.2.1. Morphological Phenotype

To elucidate the novel OxPt-R-related morphological phenotype in HCT116-OxPt-R cells, we examined morphological changes between HCT116 and HCT116-OxPt-R cells via phase-contrast microscopy after hematoxylin staining. As shown in Figure 2, the overall cell morphology was somewhat different, and the nuclear staining with hematoxylin solution was stronger in HCT116-OxPt-R cells compared to HCT116 cells (compare panels a,b). Moreover, HCT116-OxPt-R cells contained abnormal nuclear membrane structures and more aggresomes or granules compared to HCT116 cells, especially in the areas indicated by arrows (compare panels a′, b′; a″, b″; a‴, b‴). This fact suggests that certain aggresomes or granules produced in HCT116-OxPt-R cells may be a novel OxPt-R-related phenotype that may be associated with the OxPt-R property.

#### 2.2.2. OxPt-Specific Resistance

Camptothecin (CPT) is a naturally occurring pentacyclic alkaloid phytochemical (Figure 3a), exhibits selective topoisomerase I inhibition activity, and is used as a potent anticancer agent for the treatment of various cancers, including colorectal cancer [25,26]. To elucidate whether HCT116-OxPt-R cells have OxPt-specific resistance, HCT116 and HCT116-OxPt-R cells were treated with OxPt or/and CPT for 36 h, and cell viability was analyzed via CCK-8 assay. As expected, cell viability was downregulated following treatment with 5 μM OxPt in HCT116 cells (55%) but not in HCT116-OxPt-R cells (94%) (Figure 3b). However, the effect of 0.2 μM CPT on cell viability was similar in HCT116 cells (48%) and HCT116-OxPt-R cells (50%) (Figure 3c). In addition, CPT-induced cytotoxicity was enhanced by 5 μM OxPt in HCT116 cells (30%) but not in HCT116-OxPt-R cells (47%) (Figure 3c). However, HCT116-OxPt-R cells did not show significant drug resistance against high concentrations of 20 μM OxPt (Figure 3d). These results show that HCT116-OxPt-R cells have 5 μM OxPt-specific drug resistance, at least in part, but not significant resistance to 0.2 μM CPT, and a high concentration of 20 μM OxPt.

#### 2.2.3. Molecular Mechanisms

To elucidate the molecular mechanisms involved in OxPt-R, we examined apoptosis-related protein levels via Western blot analysis in HCT116 and HCT116-OxPt-R cells treated with OxPt for 36 h. According to the result, HCT116-OxPt-R cells contained upregulated p53, Noxa and γ-H2AX and downregulated caspase-9 and XIAP compared to HCT116 cells (Figure 4a,b; compare lanes 1, 5). The upregulation of p53 and Noxa and the downregulation of capase-9 and XIAP by 5 μM OxPt were significant in HCT116 cells but not in HCT116-OxPt-R cells, indicating relevance to drug resistance against 5 μM OxPt in HCT116-OxPt-R cells (Figure 4a,b; compare lanes 1, 2 and 5, 6). However, p53 and Noxa were also upregulated by 20 μM OxPt treatment in HCT116-OxPt-R cells, but not as much as in HCT116 cells, indicating partial activation of apoptotic signaling with 20 μM OxPt in HCT116-OxPt-R cells (Figure 4a; compare lanes 4, 8). Notably, γ-H2AX was upregulated by OxPt in a concentration-dependent manner in HCT116 cells, whereas upregulated γ-H2AX in HCT116-OxPt-R cells was rather downregulated by OxPt in a concentration-dependent manner (Figure 4a, third panel). Moreover, Akt, caspase-9 and XIAP were significantly downregulated by OxPt in HCT116 cells, but these proteins were not affected by 5 μM OxPt, and were somewhat upregulated by 20 μM OxPt in HCT116-OxPt-R cells (Figure 4b). Collectively, these results suggest that OxPt-R-related molecular mechanisms in HCT116-OxPt-R cells are related to the inhibition of p53-dependent apoptotic signaling.

### 2.3. Identification of New OxPt-R-Related Proteins in HCT116-OxPt-R Cells

To further elucidate OxPt-R-related molecular mechanisms, protein samples for antibody array were extracted as described in the “Sample Preparation for Antibody Array” section of “Materials and Methods” and analyzed via signaling explorer antibody array to identify new OxPt-R-related proteins. As shown in Table 1, nucleophosmin (NPM), leukocyte antigen CD37, homeobox protein Nkx-2.5, superoxide dismutase (SOD1), calreticulin, mitogen-activated protein kinase 11 (p38-β MAPK), cadherin-9, mitochondrial acyl-coenzyme A thioesterase 2 (ACOT2), high-affinity nerve growth factor receptor (TrkA), 40S ribosomal protein S27 (MPS-1), CD44 and integrin alpha-5 (ITGA5) were upregulated more than two-fold in HCT116-OxPt-R cells compared to HCT116 cells, whereas claudin-3, E3 ubiquitin-protein ligase parkin and smooth muscle gamma-actin (ACTG2) were downregulated more than two-fold in HCT116-OxPt-R cells. Moreover, acetylated histone H2B type F-S in lysine 15 (H2BS1), cleaved caspase-2 in threonine 325, cleaved matrix metalloproteinase-23 (MMP23B) in tyrosine 79, unknown lysine (Lys)-acetylated proteins and cleaved collagen alpha-1(III) chain (COL3A1) in glycine 1221 were upregulated more than two-fold in HCT116-OxPt-R cells, indicating that post-translational modifications of OxPt-R-related proteins are associated with OxPt-R properties in HCT116-OxPt-R cells. Notably, most of the OxPt-R-related proteins identified using the antibody array were upregulated in HCT116-OxPt-R cells, as shown in Figure 5, which may be related to the accumulation of certain aggresomes or granules.

### 2.4. Verification of OxPt-R-Related Proteins Identified via Antibody Array

To confirm the results in Table 1, the protein samples isolated for antibody array were analyzed via Western blot using the indicated antibodies. As expected, the protein levels of Lys-acetylated proteins, TrkA, p38 MAPK, SOD1, caspase-2, CD44 and NPM were upregulated in HCT116-OxPt-R cells comparted to HCT116 cells (Figure 6).

### 2.5. Bioinformatics Analysis for OxPt-R-Related Proteins

#### 2.5.1. Gene Ontology Analysis

The results in Table 1 were also confirmed via Western blot analysis on whole-cell extracts of HCT116 and HCT116-OxPt-R prepared using 1× SDS sample buffer. As expected, Lys-acetylated proteins, TrkA, p38 MAPK, SOD1 and caspase-2 were upregulated in the whole-cell extract of HCT116-OxPt-R compared to HCT116, whereas CD44 and NPM were somewhat downregulated in the whole-cell extract of HCT116-OxPt-R via an unknown mechanism. Thus, we analyzed the gene ontology for the newly identified OxPt-R-related proteins using the DAVID database, except for CD44 and NPM. As shown in Figure 7, the OxPt-R properties of HCT116-OxPt-R cells were significantly associated with: the positive regulation of gene expression through the upregulation of MAPK11 (p38-β MAPK), calreticulin and Nkx-2.5; the macromolecular complex through the upregulation of NTRK1 (TrkA), Nkx-2.5 and SOD1; and chaperone binding through the upregulation of calreticulin and SOD1. These results suggest that the morphological phenotype of HCT116-OxPt-R cells may be more complex due to the upregulation of OxPt-R-related proteins associated with the positive regulation of gene expression, the macromolecular complex and chaperone binding.

#### 2.5.2. Protein–Protein Interaction Network Analysis

To elucidate the OxPt-R-related protein–protein interaction network, we performed string analysis for the newly identified OxPt-R-related proteins, except for CD44 and NPM. As a result, the most significant interaction occurred between parkin (PARK2) and SOD1, suggesting that the downregulation of parkin and the upregulation of SOD1 may be associated with OxPt-R properties in HCT116-OxPt-R cells (Figure 8). In addition, the results show that protein–protein interaction networks through SOD1/p38-β MAPK (MAPK11)/TrkA (NTRK1)/calreticulin (CALR)/caspase-2 (CASP2) and MPS-1 (RPS27)/SOD1/calreticulin/Nkx-2.5/COL3A1/MMP23B could play an important role in OxPt-R-related signal transduction pathways (Figure 8).

### 2.6. Anticancer Effect by β-Lap in HCT116-OxPt-R Cells

To elucidate the anticancer effect of β-Lap in relation to OxPt-R, we investigated the effect of β-Lap on cell viability and morphological changes in HCT116 and HCT116-OxPt-R cells via CCK-8 assay and phase-contrast microscopy. As a result, the anticancer effect induced by β-Lap was significantly higher in HCT116-OxPt-R cells than in HCT116 cells, which caused more cytotoxicity and morphological changes in HCT116-OxPt-R cells (Figure 9a–c). Higher morphological changes by β-Lap in HCT116-OxPt-R cells than in HCT116 cells were demonstrated via phase-contrast microscopy after hematoxylin staining (Figure 9d).

### 2.7. Anticancer Mechanisms of β-Lap in HCT116-OxPt-R Cells

To further elucidate the anticancer mechanisms of β-Lap associated with OxPt-R, we investigated the effects of β-Lap on the regulation of apoptosis-, survival- and OxPt-R-related proteins in HCT116 and HCT116-OxPt-R cells via Western blot analysis. As shown in Figure 10a,b, p53, Noxa, γ-H2AX, cleaved caspase-8, survivin, ERK, Lys-acetylated proteins, TrkA, p38 MAPK, SOD1, caspase-2 and cleaved caspase-2 were upregulated in HCT116-OxPt-R cells compared to HCT116 cells (compare lanes 1, 5). The anticancer effect of 5 μM OxPt in HCT116 cells was related to the upregulation of p53, Noxa and cleaved caspase-8 and the downregulation of survivin, ERK, caspase-2 and CD44, whereas this phenomenon did not occur in HCT116-OxPt-R cells due to drug resistance to 5 μM OxPt (compare lanes 1, 2 and 5, 6); moreover, higher anticancer activity by β-Lap in HCT116-OxPt-R cells than in HCT116 cells was significantly associated with the downregulation of OxPt-R-related proteins, including p53, caspase-8, cleaved caspase-8, survivin, ERK, Lys-acetylated proteins, TrkA, p38 MAPK, SOD1, caspase-2, cleaved caspase-2, CD44 and NPM (lanes 5, 7, 8). These results suggest that β-Lap may play an important role in suppressing various cancer progression associated with OxPt-R through the downregulation of multiple OxPt-R-related proteins.

## 3. Discussion

Oxaliplatin (OxPt) is a widely used chemotherapeutic agent in the treatment of various cancers, including metastatic colorectal cancer. However, OxPt-induced drug resistance (OxPt-R) still remains a major issue to be solved to increase chances of successful cancer therapy. β-lapachone (β-Lap), a natural phytochemical, is expected to be developed as a chemotherapeutic agent for drug resistance caused by long-term treatment due to its potent anticancer activities [24,27,28]. In this study, we investigated the novel role of β-Lap related to oxaliplatin resistance (OxPt-R) in wild-type p53-containing HCT116 colorectal cancer cells. As a result, the anticancer activity by β-Lap was significantly higher in HCT116-OxPt-R cells resistant to 5 μM OxPt than in HCT116 cells; the anticancer mechanism of β-Lap in HCT116-OxPt-R cells was associated with the downregulation of OxPt-R-related proteins, including p53, Lys-acetylated proteins, TrkA, p38 MAPK, SOD1, caspase-2, CD44 and NPM (Figure 9 and Figure 10).

Many studies that elucidate OxPt-R-related mechanisms have been accomplished in various OxPt-R cancer cells, which are generated via long-term treatment with OxPt, with somewhat different experimental procedures depending on the cell type and researcher [17,29,30,31,32,33,34]. However, further studies on OxPt-R-related morphological properties are still essentially needed to better understand OxPt-R mechanisms. In the present study, we generated HCT116-OxPt-R cells resistant to 5 μM OxPt using our own method, as described in Figure 1b, and found that the intracellular structure of HCT116-OxPt-R is more complex than that of HCT116 cells, at least in part, due to OxPt-R-induced aggresomes or granules (Figure 1c and Figure 2). To better understand OxPt-R-related molecular mechanisms, we identified new OxPt-R-related proteins by analyzing upregulated or downregulated proteins in HCT116-OxPt-R cells compared to HCT116 cells through signaling explorer antibody array analysis. As shown in Table 1 and Figure 5, NPM1, CD37, NKX2-5, SOD1, H2BS1, CALR, MAPK11, CASP2, CDH9, MMP23B, ACOT2, COL3A1, NTRK1, RPS27, CD44 and ITGA5 were upregulated more than two-fold in HCT116-OxPt-R cells compared to HCT116 cells, whereas CLDN3, PRKN and ACTG2 were downregulated more than two-fold. In addition, gene ontology analysis of the newly identified OxPt-R-related proteins suggested that OxPt-R properties in HCT116-OxPt-R cells may be related to the positive regulation of gene expression, the macromolecular complex and chaperone binding (Figure 7). These results indicate that the upregulated OxPt-R-related proteins may be implicated in the intracellular complexity of HCT116-OxPt-R compared to HCT116 cells and the production of OxPt-R-related aggresomes in HCT116-OxPt-R cells, supporting the results of Figure 1c and Figure 2.

The tumor suppressor p53 plays an important role in DNA repair, cell cycle arrest, cell death, senescence, differentiation and metabolism in a transcriptional activity-dependent or -independent manner in response to cellular stress [35]. Colorectal cancer is one of the most common malignancies with high prevalence, and is associated with p53 mutations and impaired wild-type p53 function [36,37]. It has been shown that OxPt-induced anticancer activity was significantly higher in wild-type p53-containing colorectal cancer cells than in mutant p53 or null-type p53, whereas OxPt-R properties were higher in null-type p53-containing colorectal cancer cells than wild-type p53 [38,39], suggesting that OxPt-induced anticancer activity and OxPt-R properties may be controlled by p53 status. In the present study, we show that OxPt-R properties in HCT116-OxPt-R cells were induced in the presence of upregulated p53-mediated apoptotic signaling, as well as upregulated survivin- and ERK-mediated survival signaling (Figure 4a and Figure 10a). Moreover, our results showed that OxPt-R properties in HCT116-OxPt-R cells could be controlled by new OxPt-R-related proteins, including Lys-acetylated proteins, TrkA, p38 MAPK, caspase-2, CD44 and NPM (Figure 6). It has been shown that a single nucleotide deletion in the amino acid 382 of p53 occurred in OxPt-R cells generated from KB cells that are a subclone of human cervical carcinoma HeLa cells, and resulted in the large cytoplasmic accumulation of p53, leading to functional defects in p53 [40]. Collectively, our results suggest that OxPt-R properties in HCT116-OxPt-R cells may be related to the upregulation of functionally inactivated p53 due to probably certain mutations caused by long-term treatment with OxPt; moreover, OxPt-R properties may occur by overcoming partially activated apoptotic signaling in HCT116-OxPt-R cells through the upregulation of OxPt-R-related survival proteins.

In addition to camptothecin (CPT), β-Lap is known as a novel DNA topoisomerase I inhibitor; however, the action mode of β-Lap is different from that of CPT, and β-Lap inhibits topoisomerase I-mediated DNA cleavage induced by CPT [41,42]. β-Lap has been shown to induce apoptosis in human colon cancer cells, promyelocytic leukemia cells and prostate cancer cells, regardless of p53 status [4,5]; however, β-Lap-induced apoptosis was also associated with the activation of p53-dependent apoptotic signaling in human prostate epithelial cells through the phosphorylation of p53, the induction of Bax, and the activation of caspases [43]. In the present study, HCT116-OxPt-R cells appeared to be highly resistant to 5 μM OxPt, but not to topoisomerase I inhibitors CPT and β-Lap; cytotoxicity caused by CPT was similar between HCT116 and HCT116-OxPt-R cells, but cytotoxicity caused by β-Lap was significantly higher in HCT116-OxPt-R cells than in HCT116 cells (Figure 3 and Figure 9). In addition, our results demonstrated that p53 was greatly upregulated by 5 μM OxPt, but not by β-Lap, in HCT116 cells; moreover, the upregulated p53 in HCT116-OxPt-R cells was downregulated by β-Lap in a concentration-dependent manner (Figure 10). Since the anticancer activity of β-Lap is associated with the downregulation of mutant p53 [4,27], we strongly suggest that upregulated p53 in HCT116-OxPt-R cells may be mutated during long-term treatment with OxPt for several months, and is involved in OxPt-R properties in HCT116-OxPt-R cells.

It is known that lysine 9-acetylated histone H3 is upregulated at the MDR1 promoter in doxorubicin-resistant MCF-7 human breast cancer cells [44]; additionally, lysine 68-acetylated manganese superoxide dismutase (MnSOD) is associated with cisplatin and doxorubicin resistance due to aberrant mitochondria metabolism in MCF-7 cells [45]. In the present study, our results showed that lysine 15-acetylated histone H2B and unknown Lys-acetylated proteins were associated with OxPt-R properties in HCT116-OxPt-R cells (Table 1, Figure 6). Notably, the upregulation of Lys-acetylated proteins in HCT116-OxPt-R cells was significantly reduced by β-Lap in a concentration-dependent manner, but not in HCT116 cells, suggesting a novel anticancer mechanism of β-Lap associated with OxPt-R (Figure 10b). It is known that the dual-activation of TrkA and CD44 by NGF is involved in drug resistance to lestaurtinib in MDA-MB-231 human breast cancer cells [46]; the increase in phosphorylated p38 MAPK is associated with OxPt-R occurring in H29-D4 human colorectal cancer cells [47]; the increased caspase-2 expression in certain types of tumor has been linked to the promotion of tumorigenesis [48]; CD44 promotes resistance to etoposide-induced apoptosis in SW620 human colon cancer cells [49]; and the decreased NPM caused by trastuzumab reduces the drug resistance of gastric cancer to OxPt [50]. In the present study, we found that the downregulation of p53, survivin, ERK, Lys-acetylated proteins, TrkA, p38 MAPK, SOD1, caspase-2, CD44 and NPM by β-Lap was significantly higher in HCT116-OxPt-R cells than in HCT116 cells, and this phenomenon was associated with the higher anticancer activity of β-Lap in HCT116-OxPt-R than in HCT116 cells (Figure 9 and Figure 10).

Taken together, we generated HCT116-OxPt-R cells that are highly resistant to 5 μM OxPt from wild-type p53-containing HCT116 cells, and found that OxPt-R properties are related to certain aggresomes produced during long-term treatment with OxPt. To better understand OxPt-R-related molecular mechanisms, we identified new OxPt-R-related proteins through signaling explorer antibody array analysis, and then, compared the anticancer effect and molecular mechanisms of β-Lap between HCT116 and HCT116-OxPt-R cells. We show here that β-Lap exerts more cytotoxicity and morphological changes in HCT116-OxPt-R cells than in HCT116 cells through the downregulation of OxPt-R-related proteins such as p53, Lys-acetylated proteins, TrkA, p38 MAPK, SOD1, caspase-2, CD44 and NPM. Therefore, we propose that β-Lap could be used as an effective chemotherapeutic agent for various OxPt-R-related cancer treatments, especially in the presence of upregulated p53 and other OxPt-R-related proteins during long-term chemotherapy.

## 4. Materials and Methods

### 4.1. Materials

β-lapachone synthesized in Mazence Inc. (Suwon, Republic of Korea) was kindly provided by Prof. Gi Ryang Kweon (Chungnam National University School of Medicine, Daejeon, Republic of Korea). Oxaliplatin (Eloxatin) was obtained from Sanofi-Aventis Inc. (Seoul, Republic of Korea). Penicillin–streptomycin (10,000 U/mL) and TrypLE^TM^ Express Enzyme with phenol red were obtained from Thermo Fisher Scientific (Grand Island, NY, USA). RPMI 1640 medium was obtained from HyClone (Logan, UT, USA). Hematoxylin solution and formaldehyde solution (4%) were obtained from Merck KGaA (Darmstadt, Germany). Cell counting kit-8 (CCK-8) was obtained from Dojindo (Kumamoto, Japan). Camptothecin and phosphate buffered saline (PBS, pH 7.4) were obtained from Sigma-Aldrich (St. Louis, MO, USA). Protein assay dye reagent concentrate and 30% acrylamide/bis solution 29:1 were obtained from Bio-Rad (Hercules, CA, USA). Tween-20 and DMSO were obtained from Amresco (Solon, OH, USA). Nitrocellulose (NC) transfer membrane was obtained from GVS Life Sciences (Sanford, ME, USA). ECL Ottimo Western blot detection kit was obtained from TransLab (Daejeon, Republic of Korea). X-ray film (CP-BU NEW) was obtained from AGFA (Mortsel, Belgium). Dishes, plates, tubes and pipettes for cell culture were obtained from SPL Life Sciences (Pocheon, Republic of Korea) or Thermo Fisher Scientific (Rockford, IL, USA). p53 (DO-1), Noxa (114C307), Akt1/2/3 (H-136), Caspase-9 (96.1.23), XIAP (H-202), TrkA (763), SOD1 (FL-154), NPM/B23 (0412), caspase-8 (8CSP03), survivin (D-8), ERK1 (K-23) and GAPDH (FL-335) antibodies were obtained from Santa Cruz Biotechnology (Santa Cruz, CA, USA). Phospho-Ser139-H2AX (γ-H2AX) antibody was obtained from Upstate Biotechnology (Lake Placid, NY, USA). Antibodies against Lys-acetylated proteins and p38 MAPK were obtained from Cell Signaling Technology (Beverly, MA, USA). CD44 (EPR1013Y) was obtained from Abcam Biotechnology (Cambridge, United Kingdom). Caspase-2 antibody was obtained from BD Biosciences (San Jose, CA, USA). Secondary goat anti-rabbit and anti-mouse HRP conjugates were obtained from Bio-Rad (Hercules, CA, USA).

### 4.2. Generation of 5 μM OxPt-Resistant HCT116 Cells

HCT116 human colorectal cancer cell line was purchased from Korean Cell Line Bank (KCLB No. 10247). HCT116 cells were cultured in maintenance medium consisting of RPMI medium with L-glutamine (300 mg/L), 25 mM HEPES, 25 mM NaHCO_3_, 1% penicillin/streptomycin and 10% heat-inactivated FBS (Thermo Fisher Scientific, Grand Island, NY, USA) on a culture dish, in a 37 °C incubator supplemented with 5% CO_2_, in a humidified atmosphere. For generation of 5 μM OxPt-resistant HCT116 cells, HCT116 cells were untreated or initially treated with 0.2 μM OxPt (passage 1, P1) for 3 days in maintenance medium, and then, untreated and 0.2 μM OxPt-treated HCT116 cells were split and cultured repeatedly every 3 days. When the growth rates of untreated and 0.2 μM OxPt-treated HCT116 cells were similar, the 0.2 μM OxPt-resistant HCT116 cells were split and cultured repeatedly every 3 days in maintenance medium with sequential OxPt concentrations, as described in Figure 1b. Thus, 5 μM OxPt-resistant HCT116 cells (named HCT116-OxPt-R) were generated via long-term treatment with OxPt for several months as follows: OxPt concentration was gradually increased from 0.2 μM (P1~P3) to 0.5 μM (P1~P5), 0.6 μM (P1~P3), 0.7 μM (P1~P3), 0.8 μM (P1~P10), 0.9 μM (P1~P3) and 1.0 μM (P1~P9); greatly increased to 6.3 μM (P1~P2); reduced to 2.5 μM (P1~P10); and then, increased to 5 μM (P1~P16).

### 4.3. Phase-Contrast Microscopy

Morphology of whole cells (attached and floating cells) was analyzed via phase-contrast microscopy (EVOS XL Core, Thermo Fisher Scientific, Rockford, IL, USA) at a 10× objective (Inf Plan Achro 10× LWD PH, 0.25 NA/6.9 WD) with 150× amplification.

### 4.4. Phase-Contrast Microscopy of Hematoxylin-Stained Cells

Hematoxylin (cationic) staining is used to detect nuclei (DNA, RNA and acid nucleoprotein) [51]. Cells grown on a 6-well plate were washed with PBS, fixed with 4% formaldehyde solution, washed with PBS, and then, stained with hematoxylin solution for 24 h at RT by gently shaking. The cells were washed with PBS and were analyzed in the presence of 90% glycerol/PBS solution via phase-contrast microscopy (EVOS XL Core, Thermo Fisher Scientific) at a 20× objective (Inf Plan Fluor 20× LWD, 0.45 NA/7.1 WD) with 300× amplification.

### 4.5. Cell Viability Analysis

Cells grown on a 24-well dish were incubated with maintenance medium containing 10% CCK-8 reagent for 1.5 h in a 37 °C CO_2_ incubator. The reaction solution (100 μL each) was then transferred to a 96-well dish and was analyzed by measuring the absorbance at OD_450 nm_ using a microplate reader (SoftMax Pro 5 Software, Molecular Devices, San Jose, CA, USA).

### 4.6. Western Blot Analysis

Whole cells (attached and floating cells) were extracted with 1× SDS sample buffer and were boiled for 5 min at 95 °C. The resultant proteins were separated using SDS-PAGE and transferred to an NC membrane at 30 mA for 13–15 h. After washing with PBST (0.1% Tween-20, PBS) twice for 1 h, the membrane was blocked for 30 min at RT in blocking buffer (3% skim milk, 0.1% Tween-20, PBS), and then, incubated with primary antibody in blocking buffer at 4 °C overnight. The blot was then washed with PBST three times for 10 min, and incubated with an HRP-conjugated secondary antibody in blocking buffer for 2 h at RT. After being washed with PBST, the blot was analyzed using the ECL Western blot detection system.

### 4.7. Sample Preparation for Antibody Array

HCT116 and HCT116-OxPt-R cells were grown for 72 h and washed with PBS twice, and cell pellets were stored in a −70 °C freezer until use. To prepare protein samples for antibody array, cell pellets were extracted using protein extraction buffer (Fullmoon biosystems, Sunnyvale, CA, USA) containing 1% protease inhibitor cocktail (Sigma, St. Louis, MO, USA) and 1% phosphatase inhibitor cocktail (Sigma, St. Louis, MO, USA) and lysis beads (Fullmoon biosystems, Sunnyvale, CA, USA). After extraction, protein solution was purified using gel matrix column that was included in antibody array assay kit (Fullmoon biosystems, Sunnyvale, CA, USA). Concentration of purified sample was measured via BCA protein assay kit (Pierce, Rockford, IL, USA) using Multi-Skan FC (Thermo Fisher Scientific, Rockford, IL, USA). Additionally, purity of purified sample was confirmed on UV spectrum.

### 4.8. Signaling Explorer Antibody Array Analysis

Signaling explorer antibody array analysis result simultaneously shows the expression of 1358 proteins involved in 20 cell signaling pathways. To identify new OxPt-R-related proteins through signaling explorer antibody array analysis, 50 μg of protein sample prepared for antibody array was filled up to 75 μL with labeling buffer, and we treated 3 μL of the 10 μg/μL biotin/DMF solution at RT for 90 min with mixing. The sample was treated with 35 μL of stop reagent and incubated at RT for 30 min with mixing. The antibody microarray slide (Fullmoon biosystems, Sunnyvale, CA, USA) was treated with 30 mL of blocking solution in a Petri dish, incubated on a shaker at 60 rpm for 30 min at RT and washed with distilled water three times. The slide was rinsed with Milli-Q-grade water. The labeled sample was mixed in 6 mL of coupling solution. The blocked array slide was incubated with coupling mixture on shaker at 60 rpm for 2 h at RT in a coupling dish. The slide was washed 6 times with 30 mL of washing solution in a Petri dish on a shaker at 60 rpm for 5 min. Additionally, the slide was rinsed with Milli-Q-grade water. A total of 30 uL of 0.5 mg/mL Cy3-streptavidin (GE Healthcare, Chalfont St. Giles, UK) was mixed in 30 mL of detection buffer. The coupled array slide was treated with detection mixture in a Petri dish on a shaker at 60 rpm for 20 min at RT. The slide was washed 6 times with 30 mL of washing solution in a Petri dish on a shaker at 60 rpm for 5 min. Additionally, the slide was rinsed with Milli-Q-grade water. Slide scanning was performed using a GenePix 4100A scanner (Axon Instruments, Scottsdale, AZ, USA). The slides were completely dried before scanning and scanned within 24–48 h. The slides were scanned at a 10 μm resolution, optimal laser power and PMT. After obtaining the scan image, the scans were gridded and quantified with GenePix 7.0 Software (Axon Instruments, Scottsdale, AZ, USA). The data on protein information were annotated using UniProt DB.

### 4.9. Bioinformatics Analysis

OxPt-R-related proteins upregulated or downregulated more than 2-fold were analyzed to produce an expression heatmap and scattered image to a normalized amount (log2) using Excel-based Differentially Expressed Gene Analysis (ExDEGA) software (v.4.0.3, Ebiogen Inc., Seoul, Republic of Korea). Gene ontology analysis of OxPt-R-related proteins was performed using the Database for Annotation, Visualization and Integrated Discovery (DAVID) program (v.6.8) via the internet (http://david.ncifcrf.gov/tools.jsp) (accessed in 29 March 2023). String analysis for the protein–protein interaction network was performed using Cytoscape software (v.3.7.1.) via the internet (http://www.cytoscape.org) (accessed on 26 March 2023).

### 4.10. Statistical Analysis

Results for cell viability are presented as mean ± standard deviation of the mean. Statistical significance between control and sample was determined using Student’s *t*-test. Values of *p* < 0.05 are considered statistically significant.

## Figures and Tables

**Figure 1 ijms-24-09867-f001:**
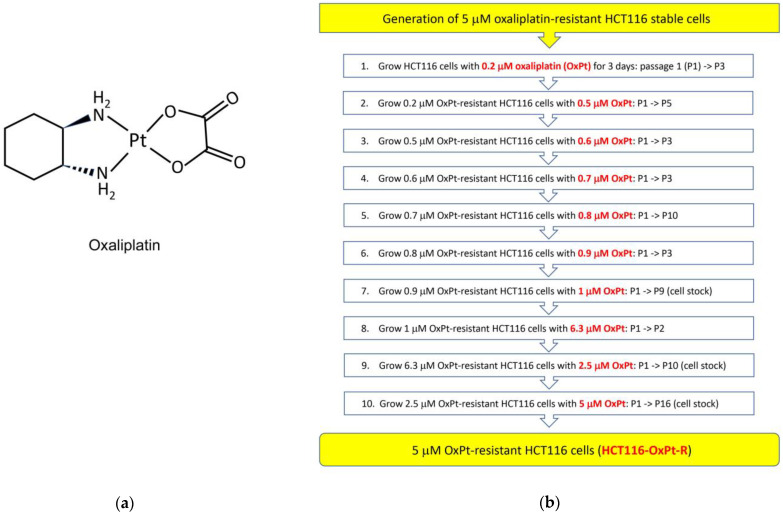
Generation of oxaliplatin-resistant HCT116 colorectal cancer cells: (**a**) Chemical structure of oxaliplatin (OxPt). (**b**) Schematic diagram of the generation of 5 μM OxPt-resistant HCT116 cells (HCT116-OxPt-R). HCT116-OxPt-R cells were generated from parental HCT116 cells by growing them in the presence of the indicated amounts of oxaliplatin every 3 days for the indicated passages. (**c**) Morphological changes between HCT116 and HCT116-OxPt-R cells. HCT116 and HCT116-OxPt-R cells from passage 17 (P17) to P22 were analyzed via phase-contrast microscopy.

**Figure 2 ijms-24-09867-f002:**
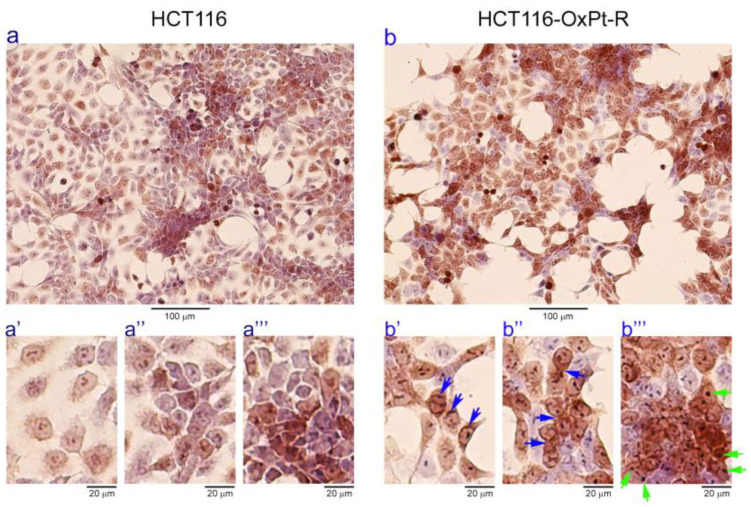
Nuclear staining of HCT116 and HCT116-OxPt-R cells with hematoxylin solution: HCT116 and HCT116-OxPt-R cells were grown for 84 h, and cell morphology was analyzed via phase-contrast microscopy after hematoxylin staining: Panels (**a′**,**a″**,**a‴**) are partly enlarged from panel (**a**); panels (**b′**,**b″**,**b‴**) are partly enlarged from panel (**b**).

**Figure 3 ijms-24-09867-f003:**
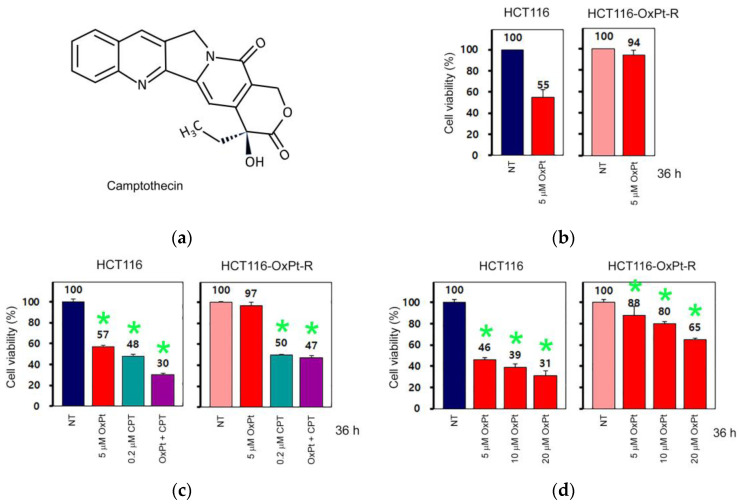
Regulation of cell viability by OxPt and camptothecin in HCT116 and HCT116-OxPt-R cells: (**a**) Chemical structure of camptothecin (CPT). (**b**–**d**) HCT116 and HCT116-OxPt-R cells were grown for 20 h, and then, treated with the indicated amounts of OxPt or/and CPT for 36 h. Cell viability was analyzed via CCK-8 assay in triplicate, either four times (**b**) or once (**c**,**d**). Error bars represent standard deviation of the mean. Statistical significance between non-treated (NT) control and drug-treated sample was determined using Student’s *t*-test, * *p* < 0.05 (**c**,**d**).

**Figure 4 ijms-24-09867-f004:**
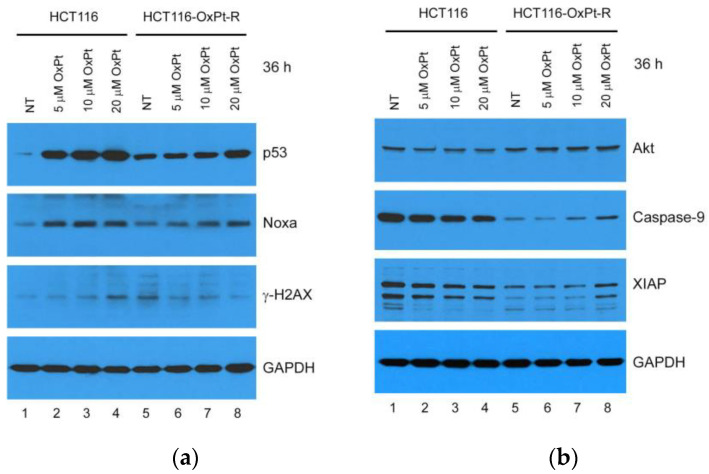
Regulation of apoptosis-related proteins by OxPt in HCT116 and HCT116-OxPt-R cells: (**a**,**b**) HCT116 and HCT116-OxPt-R cells were grown for 20 h, and then, treated with the indicated amounts of OxPt for 36 h. Whole-cell extracts were prepared using 1× SDS sample buffer and analyzed via Western blot using the indicated antibodies.

**Figure 5 ijms-24-09867-f005:**
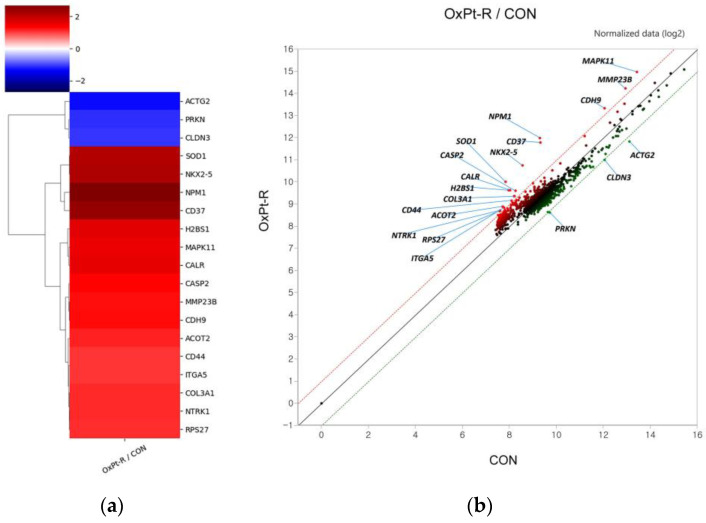
Expression heatmap and scattered image for OxPt-R-related proteins in HCT116-OxPt-R cells: (**a**) Expression heatmap image for proteins altered more than 2-fold. (**b**) Scattered image for normalized expression amount. Upregulated proteins are shown in red with their gene symbols, and downregulated proteins are shown in blue (**a**) or green (**b**) with their gene symbols.

**Figure 6 ijms-24-09867-f006:**
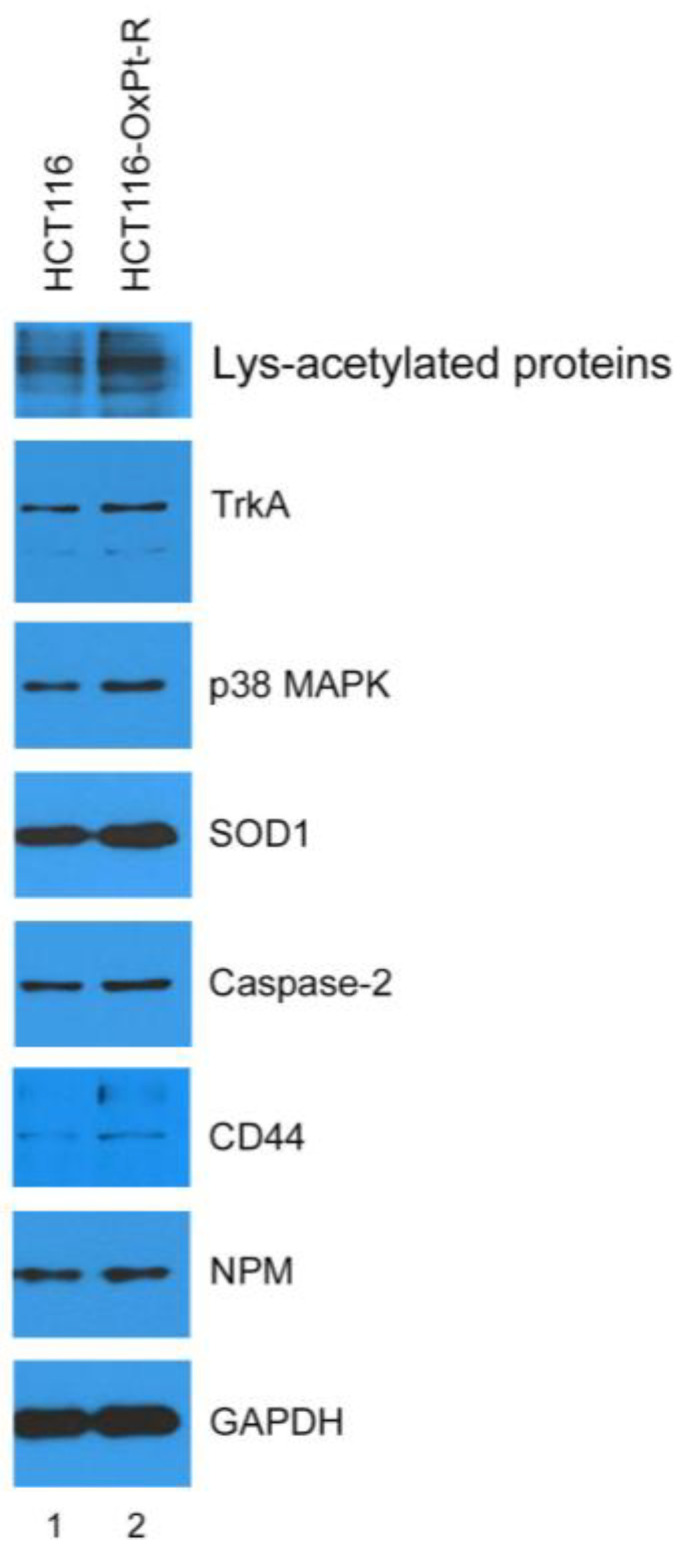
Verification of OxPt-R-related proteins identified via antibody array: HCT116 and HCT116-OxPt-R cells were grown for 72 h, and protein samples for antibody array were prepared as described in “Materials and Methods”, and then, analyzed via Western blot using the indicated antibodies.

**Figure 7 ijms-24-09867-f007:**
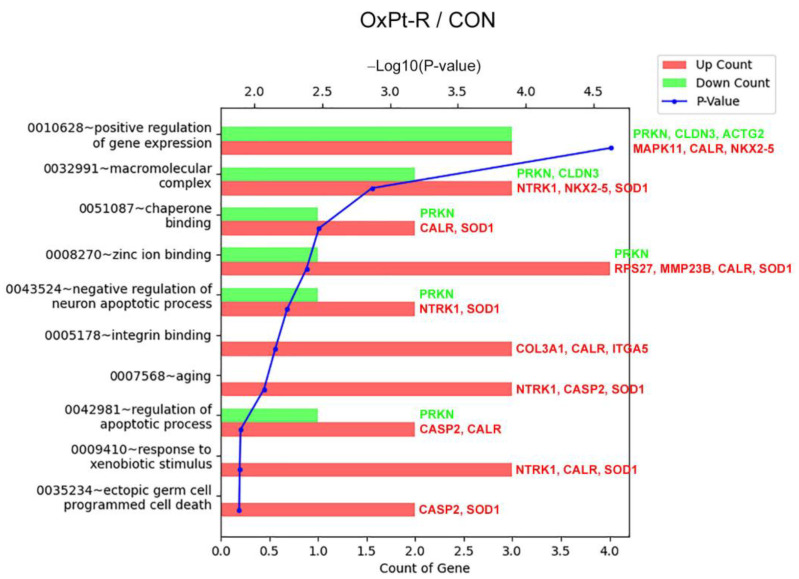
Gene ontology analysis of OxPt-R-related proteins using the DAVID database: upregulated and downregulated proteins are shown in red and green, respectively, with their gene symbols.

**Figure 8 ijms-24-09867-f008:**
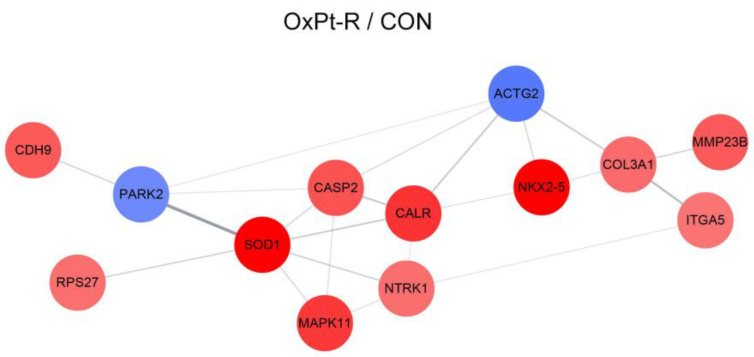
String analysis of OxPt-R-related proteins: Protein–protein interaction networks were proposed via string analysis using Cytoscape software (v.3.7.1.). Upregulated and downregulated proteins are shown in red and blue, respectively, with their gene symbols. PARK2 is a synonym for PRKN.

**Figure 9 ijms-24-09867-f009:**
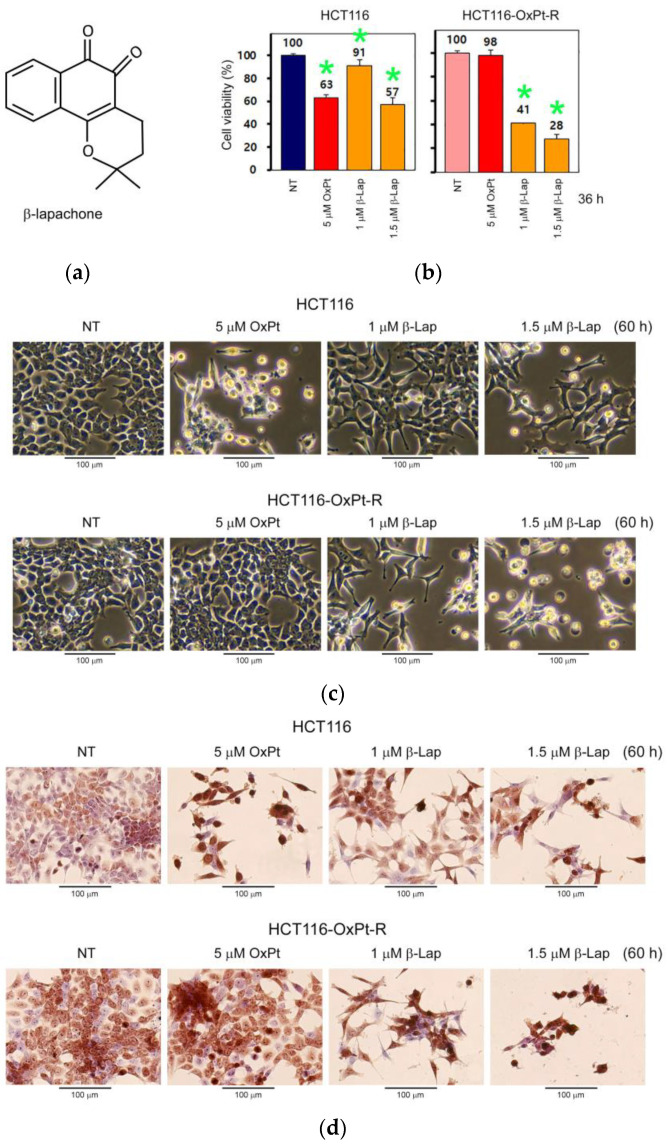
Regulation of cell viability and morphology by β-Lap in HCT116 and HCT116-OxPt-R cells: HCT116 and HCT116-OxPt-R cells were grown for 20 h, and then, treated with OxPt or β-Lap for the indicated amount and time. (**a**) Chemical structure of β-lapachone (β-Lap). (**b**) Cell viability was analyzed via CCK-8 assay in triplicate. Statistical significance was determined using Student’s *t*-test, * *p* < 0.05. (**c**) Phase-contrast microscopy. (**d**) Phase-contrast microscopy after hematoxylin staining.

**Figure 10 ijms-24-09867-f010:**
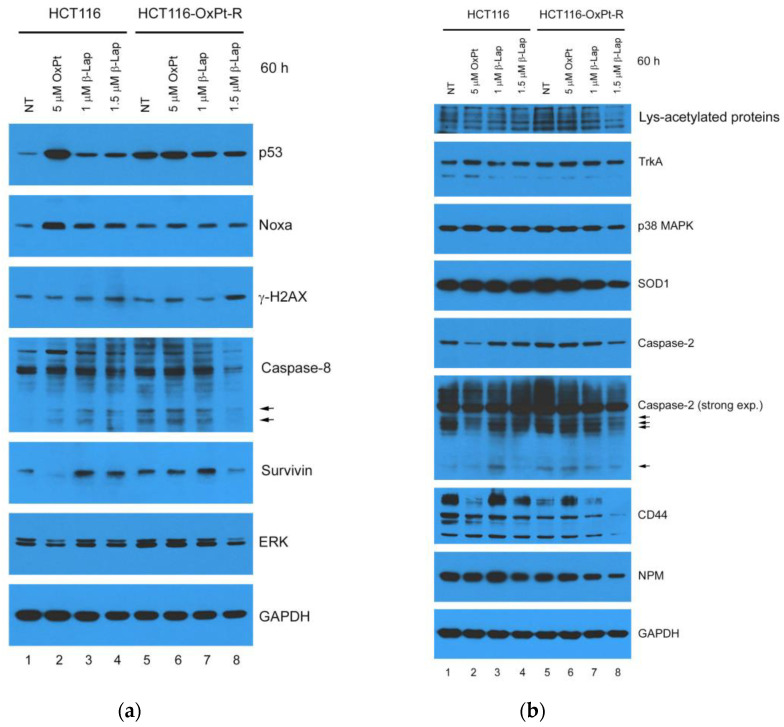
Regulation of OxPt-R-related proteins by β-Lap in HCT116 and HCT116-OxPt-R cells: (**a**,**b**) HCT116 and HCT116-OxPt-R cells were grown for 20 h, and then, treated with the indicated amounts of OxPt or β-Lap for 60 h. Whole-cell extracts were prepared using 1× SDS sample buffer and analyzed via Western blot using the indicated antibodies.

**Table 1 ijms-24-09867-t001:** Identification of new OxPt-R-related proteins in HCT116-OxPt-R cells through signaling explorer antibody array.

Gene Symbol	Antibody Name	Protein Name	Fold Change OxPt-R/CON	UniProt ID
NPM1	Nucleophosmin (NPM)	Nucleophosmin	6.433	P06748
CD37	CD37	Leukocyte antigen CD37	5.477	P11049
NKX2-5	NKX2.5	Homeobox protein Nkx-2.5	4.541	P52952
SOD1	SOD1	Superoxide dismutase [Cu-Zn]	4.460	P00441
H2BS1	Histone H2B (Acetyl-Lys15)	Histone H2B type F-S	3.147	P57053
CALR	Calreticulin	Calreticulin	3.017	P27797
MAPK11	MAPK 11	Mitogen-activated protein kinase 11	2.925	Q15759
CASP2	CASP2 (p18, Cleaved-Thr325)	Caspase-2	2.518	P42575
CDH9	CDH9	Cadherin-9	2.434	Q9ULB4
MMP23B	MMP23 (Cleaved-Tyr79)	Matrix metalloproteinase-23	2.413	O75900
ACOT2	ACOT2	Acyl-coenzyme A thioesterase 2, mitochondrial	2.256	P49753
N/A	Lys-acetylated proteins	Lys-acetylated proteins	2.241	N/A
COL3A1	Collagen III alpha1 (Cleaved-Gly1221)	Collagen alpha-1(III) chain	2.188	P02461
NTRK1	TrkA	High-affinity nerve growth factor receptor	2.162	P04629
RPS27	MPS-1	40S ribosomal protein S27	2.144	P42677
CD44	CD44	CD44 antigen	2.090	P16070
ITGA5	Integrin alpha-5 (ITGA5)	Integrin alpha-5	2.084	P08648
CLDN3	Claudin 3	Claudin-3	0.480	O15551
PRKN	Parkin	E3 ubiquitin-protein ligase parkin	0.465	O60260
ACTG2	Actin-gamma2	Actin, gamma-enteric smooth muscle	0.411	P63267

## Data Availability

Not applicable.

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
