# Peer review of "β-Lapachone Exerts Anticancer Effects by Downregulating p53, Lys-Acetylated Proteins, TrkA, p38 MAPK, SOD1, Caspase-2, CD44 and NPM in Oxaliplatin-Resistant HCT116 Colorectal Cancer Cells"

_ijms, 2023, doi:10.3390/ijms24129867_

Round 1

Reviewer 1 Report

Colorectal cancer is one of the most common malignant cancers worldwide. In this article, the authors first generated and characterized  mM OxPt-resistant HCT116 cells (HCT116-OxPt-R) to reveal a novel role of b-Lap associated with OxPt-R in wild-type p53-containing HCT116 cells. They compared the anticancer effect and molecular mechanism by b-Lap between HCT116 cells and HCT116-OxPt-R cells. As a result, the anticancer effect by b-Lap was higher in HCT116-OxPt-R cells than in HCT116 cells, leading to more cytotoxicity and morphological changes in HCT116-OxPt-R cells.  The anticancer mechanism by b-Lap in HCT116-OxPt-R cells containing certain aggresomes and upregulated p53 was associated with downregulation of OxPt-R-related proteins such as p53, survivin, ERK, Lys-acetylated proteins, TrkA, p38 MAPK, SOD1, caspase-2, CD44 and NPM. All this determines the relevance of the problem under consideration and the significance of the results presented by the authors. The literature cited in this paper is sufficient in volume and content. All 10 figures are of sufficiently high quality. I recommend that the journal editors accept the article with minor technically corrections .

This study is very interesting and useful.

Dear Author!

 Please revisit the article with reg. no.2434200  in accordance with Editor’s comments:

·         In line 104-298 at point 2. Results (2.1-2.7.) the text must be double-aligned.

·         General conclusions are missed. Must be added!

Reviewer 2 Report

Submissted manuscript, "Beta-lapachone Exerts Anticancer Effects by Downregulating p53, Lys-Acetylated Proteins, TrkA, p38 MAPK, SOD1, Caspase-2, CD44 and NPM in Oxaliplatin-Resistant HCT116 Colorectal Cancer Cells". There are various typographical errors, and the following points must be addressed.

1. Add a 2D structure of Beta-lapachone, Camptothecin and Oxaliplatin, to improve the readability of the paper.

2. Rationality behind choosing Oxaliplatin-resistant cells to evaluate the activity of beta-lapachone, when conceptually they belong to different classes of structures (or drug classes). 

3. Please provide sufficient justification for choosing HCT-116 cells. 

4. Please clearly state the standards or controls used during these experiments; seems a bit confusing.

5. In Table 1, have the authors found any correlation between these proteins?

The Authors need to correct the English, while some sentences need scientific  correction.
